# Au Wire Ball Welding and Its Reliability Test for High-Temperature Environment

**DOI:** 10.3390/mi13101603

**Published:** 2022-09-27

**Authors:** Chenyang Wu, Junqiang Wang, Xiaofei Liu, Mengwei Li, Zehua Zhu, Yue Qi

**Affiliations:** 1Notional Key Laboratory of Instrumentation Science & Dynamic Measurement, Taiyuan 030051, China; 2Academy for Advanced Interdisciplinary Research, North University of China, Taiyuan 030051, China

**Keywords:** Au wire bonding, high-temperature annealing, focused ion beam (FIB), morphology analysis

## Abstract

The long-term application of sensors in a high-temperature environment needs to address several challenges, such as stability at high temperatures for a long time, better wiring interconnection of sensors, and reliable and steady connection of the sensor and its external equipment. In order to systematically investigate the reliability of thin coatings at high temperatures for a long time, Au and Cr layers were deposited on silicon substrates by magnetron sputtering. Additionally, samples with different electrode thicknesses were annealed at different temperatures for a varied duration to study the effect of electrode thickness, temperature, and duration on the reliability of samples. The results of tensile and probe tests before and after heat treatment revealed that the mechanical strength and electrical properties have changed after annealing. In addition, the bonding interface was analyzed by a cross-sectional electron microscope. The analysis showed that long-term continuous high-temperature exposure would result in thinning of the electrode, formation of pores, recrystallization, and grain growth, all of which can affect the mechanical strength and electrical properties. In addition, it was observed that increasing the thickness of the gold layer will improve reliability, and the test results show that although the thin metal layer sample is in poor condition, it is still usable. The present study provides theoretical support for the application of thin coatings in high temperatures and harsh environments.

## 1. Introduction

Transducers, such as temperature sensors and high-temperature pressure sensors, are used in harsh environments, including aerospace engines, supersonic aircraft, automotive engines, and metallurgical industries [1,2,3,4,5]. These sensors are used for quite a long time in harsh environments [6,7,8]. One of the most critical technologies in sensor packaging is wire bonding, wherein the bonding materials and bonding parameters will affect the quality and operation life of sensors [9]. Therefore, it is necessary to analyze the long-term stability, service life, ability to withstand harsh environments, mechanical strength of wire bonding. Thus, these bottlenecks need to be addressed urgently in developing a suitable long-term stable contact and packaging technology [10,11].

High purity Au wires have been used as an interconnection for most device packages [12,13] as Au has high temperature resistance, excellent conductivity, and good oxidation resistance. Presently, gold wire ball welding has become one of the commonly used welding processes in sensor packaging technology [14,15]. In general, Au wires ball welding is produced by the electric-flame-off (EFO) system, which generates high-voltage discharge to the tail wire and generates electric sparks. The high temperature generated instantly melts the tail end of the gold wire to produce a free air ball (FAB) [16,17]. Later, the ultrasonic power source generates mechanical vibration energy, which causes deformation between the gold ball and the pad, resulting in mutual diffusion of metal atoms. This completes the welding of the first solder joint; generally, the height of the welding ball welded by this process is 2–3 times the wire diameter. The factors that affect the quality of welding points are ultrasonic power, contact time, welding pressure, and hot table temperature [18,19,20,21,22], all of which need to be adjusted to maximize the welding quality.

Generally, the barrier layer will be sputtered under the Au layer to increase the adhesion with the substrate and prevent the Au layer from diffusing between the substrates. The most common metals used for barrier layers are Ti, Cr, Ni, and Ta [23,24,25,26]. The traditional pad coating is composed of 3–5 µm nickel layer and 1–1.25 µm thick gold layer. The processing difficulty and high cost of gold, thinner coating such as 0.3–0.5 µm of the gold layer are opted to replace the thick gold layer. Nevertheless, the thin coating of gold leads to some challenges, including poor bonding of lead wire, especially after high-temperature baking, and failure of bond due to the lower annealing temperature of Au (600 °C) [27,28,29,30,31]. In practical application, high-temperature sensors need to work in high-temperature and harsh environments. Therefore, it is necessary to analyze the lead bonding failure at high temperatures and explore the bonding failure mechanism at different thicknesses and temperatures in order to develop better bonding strategies.

In this work, the Au wire ball bonding method is employed. Under the optimal parameters debugged, Au wire ball bonding has been carried out to produce a thin metal coating. The bonding quality has been analyzed by testing the pull force of gold wire on the coatings with different thicknesses. The thin metal coating of 50/300 nm and 50/400 nm of Cr/Au plated on bare silicon by magnetron sputtering, separately. Then the temperature resistance experiments were carried out at 600 °C and 700 °C for 2 h and 5 h, separately. Further, the test samples were analyzed by scanning electron microscopy (SEM) and probe station, and the morphology and electrical conductivity of the samples were observed, and their stability under the harsh environment at high temperature for a long time was assessed.

## 2. Materials and Methods

The investigated Au wires with 99.99% purity and 25 µm diameter were bonded to the gold electrodes on the silicon substrate by ball bonding in a wiring machine. Figure 1 shows the process of fabricating a gold electrode on a 2-inch diameter bare silicon chip. In Figure 1a, bare silicon wafers are treated in a vacuum with oxygen plasma to remove surface contaminants and enhance the adhesion between the wafer and the sputtered metal. After photolithographic treatment of bare silicon wafers, as shown in Figure 1b, a 50-nm layer of metal Cr is first sputtered on the surface of the wafer using a magnetron sputtering machine (LAB18, K. J. Lesker, Inc., Jefferson Hills, PA, USA) as a seed layer. Afterward, 300-nm and 400-nm thicknesses of Au are sputtered on the surface of the wafer as bonded electrodes. Subsequently, the samples soaked in acetone solution were stripped to remove excess Cr/Au metals. The stripping results are shown in Figure 1c. Finally, the prepared sample was diced as the final experimental slice Figure 1d. The sample is square, the size of the square is 1 cm × 1 cm, and the electrode spacing is 1 mm after dicing according to the above process steps. Each sample contains 9 pairs of electrodes with opposite directions.

The wire bonding process was performed on the two electrodes of the cut experimental sample (Micro Point Pro, iBond5000, Yokneam Elite, Israel). During the bonding process, the ultrasonic and heat energies provided by the cleaver were transferred to the welding wire and pad. The energy absorbed by gold wire is
(1)$$E=k\Delta T^{2}+uFAft$$
where *k* is the influence coefficient, *ΔT* is the temperature difference before and after wire bonding, *u* is the friction coefficient, *F* is the pressure exerted by the cleaver, *A* is ultrasonic amplitude, f is ultrasonic frequency, and *t* is the time of ultrasonic action on the gold wire. There are several key factors affecting the quality of solder joints. During the bonding process, we carried out a series of optimizations and finally determined the best bonding pressure as 0.4 MPa, corresponding to the parameter *F* in the above formula; the ultrasonic power as 100 mV. The ultrasonic power is the ultrasonic (U/s) energy applied to the welding spot. The ultrasonic power is divided into two modes: high power and low power. The parameters of high power correspond to large ultrasonic amplitude. The two parameters in the above formula, ultrasonic amplitude A and ultrasonic frequency f, are reflected by the ultrasonic power; the ultrasonic time as 30 ms, which corresponds to the parameter t in the above formula; and the hot bench temperature as 150 °C, corresponding to the temperature difference *ΔT* before and after wire bonding in the above formula. In addition, the looping parameters are loop top, reverse length, and kink height, all of which have an impact on the overall bonding quality, and the functional relationship between them is shown in Figure 2. The diameter of the ball is generally 2–5 times of the diameter of the gold wire, the height of the loop top is generally between 200 μm and 800 μm, which can be adjusted according to the actual application, the reverse is to ensure that the gold wire is arcuate. In this paper, the loop top set in the wire bonding machine is 600 μm, the kink height is 300 μm, and the reverse length is 50 μm.

The sample characteristics and experimental conditions are shown in Table 1. The Cr/Au metal layer with a thickness of 50/300 nm and 50/400 nm is maintained in the high-temperature environment of 500 °C and 700 °C for 2 h and 5 h, respectively. Then the electronic scanning mirror and probe table resistance of the samples are observed after high-temperature annealing, and the energy of the thin metal layer to withstand high temperature, and harsh environment is obtained.

## 3. Results and Discussion

### 3.1. Bonding Result and Quality Test

After gold wire ball bonding, the bonding samples were observed under an optical microscope and scanning electron microscope to identify the bonding bumps, lead arc, and overall bonding morphology. Figure 3a,b shows the observation results under the optical microscope and the scanning electron microscope. Figure 3c shows the bonding sphere and the position of the electrode surface under a high-power microscope. It can be observed from the SEM image that the shape of the solder ball is three-dimensional, the length of the tail line generated by translation is moderate, and no dragging phenomenon was identified, which ensures the reliability of the bonding results. In addition, the thickness of the electrode Pad was measured with a step meter to ensure the consistency of the experiment. Figure 3d,e show that the thickness of the two electrodes, as expected, was found to be 367.9nm and 482.2 nm. Further, the horizontal curve confirms the uniform thickness of the film. Thus, the quality and uniformity of the metal film are achieved.

Further, the destructive tensile test was performed on the bonded samples to assess the stability. The test requirements must meet two standards. Firstly, the measured pull force must be equal to or greater than 50% of the maximum yield force of the undeformed wire. In addition, the standard deviation of the sample in the laboratory should be less than 15%, and the standard deviation of the sample for manufacturing application should be less than 25%. Another prerequisite is that there is no bonding layer in laboratory applications and less than 10% in manufacturing applications [32]. The pull test results are shown in Table 2, and the data is in agreement with both criteria. The destructive tensile test data suggests that the critical failure force of the sample with a 400 nm thick electrode gold layer is greater than that of the 300 nm thick gold layer. Therefore, the metal layer with an appropriate thickness within the acceptable range can enhance the bonding quality of the sample.

### 3.2. Testing of Mechanical Properties before High Temperature Annealing

Further, the destructive tensile test was performed on the bonded samples to assess the stability. The test requirements must meet two standards. Firstly, the measured pull force must be equal to or greater than 50% of the maximum yield force of the undeformed wire. In addition, the standard deviation of the sample in the laboratory should be less than 15%, and the standard deviation of the sample for manufacturing application should be less than 25%. Another prerequisite is that there is no bonding layer in laboratory applications and less than 10% in manufacturing applications [32]. The pull test results are shown in Table 2, and the data is in agreement with both criteria. The destructive tensile test data suggests that the critical failure force of the sample with a 400 nm thick electrode gold layer is greater than that of the 300 nm thick gold layer. Therefore, the metal layer with an appropriate thickness within the acceptable range can enhance the bonding quality of the sample.

The hot press ball welding generally involves two solders, the first solder joint is ball welding, and the second solder joint is wedge welding, in short, ball-wedge bonding mode. As the bonding tension of ball welding is much greater than wedge welding bonding, it is necessary to carry out reinforcement treatment in the second solder joint. For materials with small breaking forces, such as gold wire, the ball pressing reinforcement method can be used for repair welding.

The equipment used in the experiment is the Dage4000 push-pull test system. The measurable pull range is 1 g–10,000 g, the system accuracy is ±0.25%, and the maximum test speed can reach 5000 μm/s. The experimental method used in this experiment is to place a pull hook directly below the middle position of the lead and slowly pull it in the *z*-axis direction until the lead breaks. The moving speed is set at 2500 μm/s, and the accuracy of the *z*-axis is ± 10 μm. As shown in Figure 4, both solder joints are spherical. Figure 4a,b show an optical microscope picture and the SEM image of the fracture position after the tensile failure test, respectively. As can be observed from the figure, the fracture position is concentrated in the spherical tail of the first and second solder joints and does not appear at the second solder joint, indicating that the ball pressing reinforcement has a significant effect on the bonding strength of the second solder joint.

In addition, the shear force test was also carried out. The average shear force of the electrode with a thickness of 300 nm was 7.506 g, and the average shear force of the electrode with a thickness of 400 nm was 12.825 g. Figure 4c shows the shear force test curve of one point. Cracking occurred at the maximum of 8.091 g. The test process was conducted according to JESD22-B117 standard. Figure 4c is a picture of one of the experimental data.

### 3.3. High-Temperature Reliability Test

In order to test the bonding reliability of samples under high temperature and harsh environments and test the performance of thin metal layer after high-temperature annealing, 10 samples with different metal layer thicknesses were randomly selected for the high-temperature annealing experiment. Similarly, the samples from the same batch were also selected and placed at room temperature for the same time for comparative experiments. The selected samples were heated in a high-temperature oven for 2 and 5 h, and the annealing is carried out in an atmospheric atmosphere using a tubular annealing boiler separately, and heated cross sections were prepared for comparison with samples at room temperature. The samples after high-temperature annealing and the control group samples were observed and compared under the scanning electron microscope. The results are shown in Figure 5. Initially, the surface of the sample was observed, followed by the changes in the morphology of the bonding interface before and after annealing by cutting the bonding interface. In general, the cutting can be carried out by using the grinding prototype to solidify the silicone grease grinding sample, or by using a focused ion beam (FIB). The present work used the FIB method.

The samples before and after annealing are cut, and the profile is observed. It can be clearly observed from Figure 5a that before annealing, the Au and Cr layer of the sample have obvious interfaces, and the section is flat without any cavities and cracks. Moreover, the thickness of the Au/Cr layer (51.37/350.9 nm) measured by a large magnification electron microscope is consistent with the actual measured value. Figure 5b shows the cross-section at the electrode. The situation at the electrode is similar to that at the bond. The Au and Cr layers are flat and smooth. Figure 5c,d are SEM images of 300-nm-thick electrodes annealed at 500 °C for 2 h and 5 h, separately. The joints of the electrodes and the balls are not cracked, while there are many cracks and holes in the electrodes. However, the detailed deeper impact can be studied by observing the cut profile. Figure 5e shows the FIB cut interface between the bonded free gold balloon and the electrodes after annealing for 5 h at 700 °C. The high-power electron microscopy of the cut interface confirms that there is no longer a gap between the bonded free gold balloon and the electrodes. This is because the grain size of the bonded free gold balloon increases with temperature, disappearing the gap. However, some cracks and holes are still observed. This suggests the positive effect of heat treatment on the bonding interface between the gold sphere and the electrodes.

Figure 5f shows the image of a 300 nm thick electrode annealed at 500 °C for 2 h. From the figure, it can be observed that a small part of the gold layer evaporated, while most of it is still intact, and only some small holes appeared, without obvious fracture. Figure 5g shows the image of a 300 nm thick electrode annealed at 500 °C for 5 h. The gold layer has become significantly thinner, accompanied by some large extended gaps. Notably, some crystalline grains have also been observed, indicating that recrystallization and grain growth have occurred, and this can well explain the disappearance of gaps in Figure 5e. The 300-nm thick gold electrode samples annealed for 2 and 5 h at 700 °C are similar to those annealed for 2 and 5 h at 500 °C, except that more voids and more severe grain regeneration were observed at 700 °C, but the difference is not significant from that at 500 °C, so it is not discussed here.

Figure 5h is the image of 400-nm thick electrodes annealed at 500 °C for 5 h. The image of 2 h annealing is also similar to that of the 300 nm thick electrode. Unlike 300 nm thick electrodes with voids, there are no obvious voids. However, there are some small cracks and voids, and the remaining thickness of the electrodes is consistently thick. Therefore, it can be concluded that increasing the thickness of electrodes can improve the high-temperature resistance of samples. Figure 5i shows the image of a 400 nm thick sample after annealing at 700 °C for 2 h. Compared with Figure 5f, it can be clearly seen that although there are some voids, the homogeneity of the gold layer has been significantly improved. Moreover, increasing the thickness of the gold layer has a positive impact on the temperature resistance of the sample. Figure 5j is the image of a 400 nm thick sample after annealing at 700 °C for 5 h. In addition to obvious voids and grain growth, there are some similar faults identified in the gold layer. The thickness of the gold layer can be clearly seen, and the resistance is also identified as normal when tested with the probe bench. This suggests that the 400 nm thick sample can work for a long time in the high-temperature environment of 700 °C.

It can be concluded that both 50/300 nm and 50/400 nm thick Cr/Au electrodes can withstand high temperatures and harsh environments for a long time. Within the allowable range, the thicker the electrode, the higher the tolerable temperature, and the longer the temperature resistance time. The high-temperature environment has a significant positive impact on the interface between free air golden ball and electrode bonding.

### 3.4. Comparative Testing of Mechanical and Electrical Performance

Destructive tensile tests were carried out on several groups of samples after annealing and compared with that before annealing. The results are shown in the histogram in Figure 6. It can be observed that the tensile strength of the sample after high-temperature annealing is biased to a lower value. After annealing for 2 h, the pull value of the samples decreased by nearly 50%, and the samples treated at 700 °C showed a larger decrease than that treated at 500 °C. The critical tensile failure value of the samples annealed at high temperature for 5 h is lower than 1/4 of that of the samples not annealed. Due to the evaporation of the gold wire in the high-temperature environment, the diameter decreases, the mechanical strength of the gold wire decreases, and thus the strength of the connection between the free air gold ball and the electrode also decreases. The reason for the reduction of the tension value may be attributed to these two factors.

The resistance values of samples before and after high-temperature annealing were measured with a probe table. As can be observed from the histogram, the resistance values of 300 nm and 400 nm thick samples after annealing at 500 °C and 700 °C for 2 h have decreased. The range of resistance reduction is as small as 15%, which can be attributed to the evaporation of electrode and gold wire at high temperatures. However, the resistance values of the two samples showed an increase by nearly twice the original value after annealing at 500 °C for 5 h. This increase can be attributed to various factors such as the diffusion of Cr layer metal to the Au layer, the diffusion of oxygen atoms to the Au layer, and the occurrence of recrystallization and grain growth. In addition, an interesting phenomenon is that the resistance of samples annealed at 700 °C for 5 h shows a decreasing trend. This is probably due to the different diffusion degrees of Cr and O atoms at different temperatures, and the elimination of metal dislocation and stress by heat treatment [33].

In this work, the resistivity of the annealed film can be expressed as [34]:(2)$$\rho =\rho_{Au}+\rho_{CrI}+\rho_{OI}+\rho_{D}$$
where $\rho_{Au}$ is the Au bulk resistivity, $\rho_{CrI}$ is the resistivity due to diffusion of Cr atoms into the Au layer, $\rho_{OI}$ is the resistivity due to diffusion of O atoms into Au layer, and $\rho_{D}$ is the resistivity caused by impurities, point defects, and grain boundaries in the Au layer.

For the Au electrode annealed at 500 °C and 700 °C, as discussed in Figure 7, high temperature will produce grains, recrystallization and impurities, which can be considered that $\rho_{D}$ in the above formula decreases. The conductivity of Au is better than that of Cr, a higher temperature will cause the Cr layer to diffuse into the Au layer, and the O atoms in the air will also diffuse into the Au layer, which will lead to an increase in $\rho_{CrI}+\rho_{OI}$. There is only slight grain Recrystallization at 500 ℃, so the main reason for the increase in resistance at 500 ℃ is that Cr and O atoms diffuse into the Au layer, resulting in an increase in resistance. The diffusivity of Cr and O atoms in Au is higher at 500 ℃ than at 700 ℃, and recrystallization is more obvious at 700 ℃. At the same time, heat treatment can also eliminate stress and dislocation in the metal. As a result, the resistivity at 700 ℃ is much lower than that at 500 °C [34].

## 4. Conclusions

The present work mainly emphasizes the thermoelectric reliability of a gold wire ball welding system based on a silicon substrate. Samples of Cr/Au electrodes with electrode thickness of 50/300 nm and 50/400 nm were annealed at high temperatures of 500 °C and 700 °C for 2 h and 5 h, respectively, to simulate the high temperature and harsh environment in practical applications. The results showed that after high-temperature annealing, the gap between the gold ball and the electrode bonding interface disappears, and high-temperature heat treatment has a positive effect on the bonding interface. However, as the annealing temperature and time increase, some pores will appear, accompanied by recrystallization and grain growth, affecting the electrical and mechanical properties of the sample.

In a comparison of the annealed samples after the tensile destructive test and the samples at room temperature, it is found that the critical value of tensile force is significantly reduced, indicating that high-temperature heat treatment significantly reduces the mechanical strength of the samples. The resistance values of annealed samples were measured with a probe station. It was found that when the samples were treated at a high temperature for a short time, the resistance values decreased due to the thermal volatilization of gold. However, when the samples were annealed at 500 °C for a long time, the resistance of the samples increased due to the diffusion of Cr and O atoms and the growth of grains. As the activation free energy of gold decreases at 700 °C and heat treatment eliminates metal dislocation and stress, the resistance value of samples after heat treatment for a long time at 700 °C decreases significantly below the initial value. Conclusively, the present work can provide the basis for the selection of thin metal layers for gold wire ball bonding and improve the reliability of application in high temperature and harsh environments to promote its application in the field of high-temperature pressure sensors or high-temperature sensors.

## Figures and Tables

**Figure 1 micromachines-13-01603-f001:**
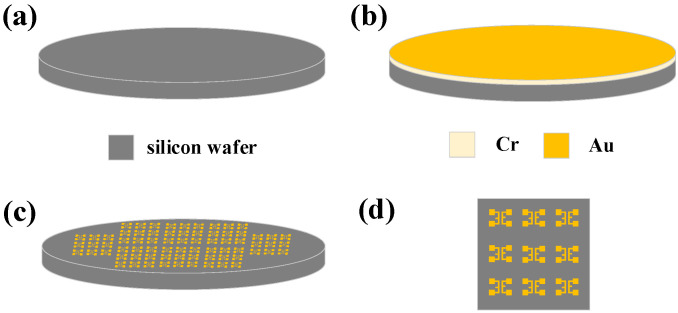
Process flow diagram for fabricating Cr/Au electrodes on silicon wafers. (**a**) Bare silicon pre-treatment. (**b**) Sputtering Cr/Au metal layer. (**c**) Acetone immersion peeling. (**d**) Dicing Silicon Crystals.

**Figure 2 micromachines-13-01603-f002:**
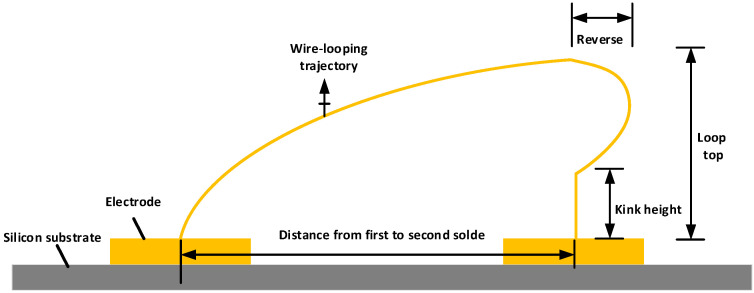
3D schematic diagram of key parameters of gold wire ball welding bonding.

**Figure 3 micromachines-13-01603-f003:**
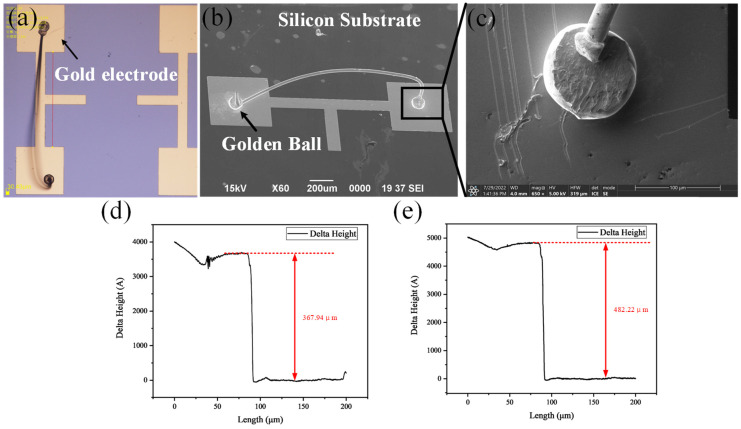
Analysis of interface and electrodes after ball bonding. (**a**) Photo of electrodes and integrated bonding optical microscopy. (**b**) Scanning Electron Microscope Images of Electrodes and Global Bonding. (**c**) Interface Diagram of Spherical Welding under High Power Scanning Electron Microscope. (**d**) Measuring 50/300 nm thick Cr/Au electrodes with a step instrument. (**e**) Measuring 50/400 nm thick Cr/Au electrodes with a step instrument.

**Figure 4 micromachines-13-01603-f004:**
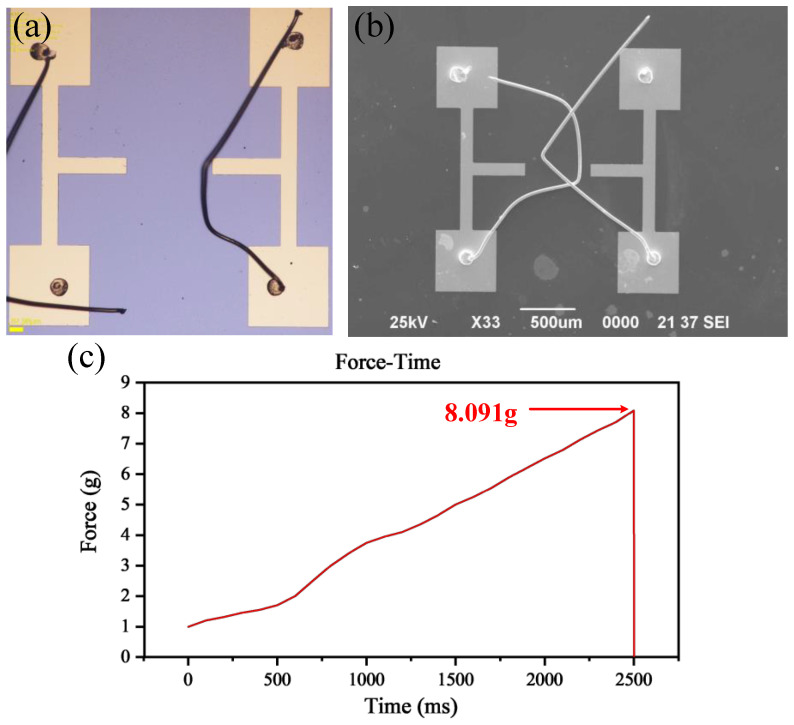
Pull failure experimental results observed under the optical microscope and scanning electron microscope and photo of shear force test. (**a**) Optical microscope image after tensile test. (**b**) SEM image after tensile failure test. (**c**) Time-force diagram of shear force test.

**Figure 5 micromachines-13-01603-f005:**
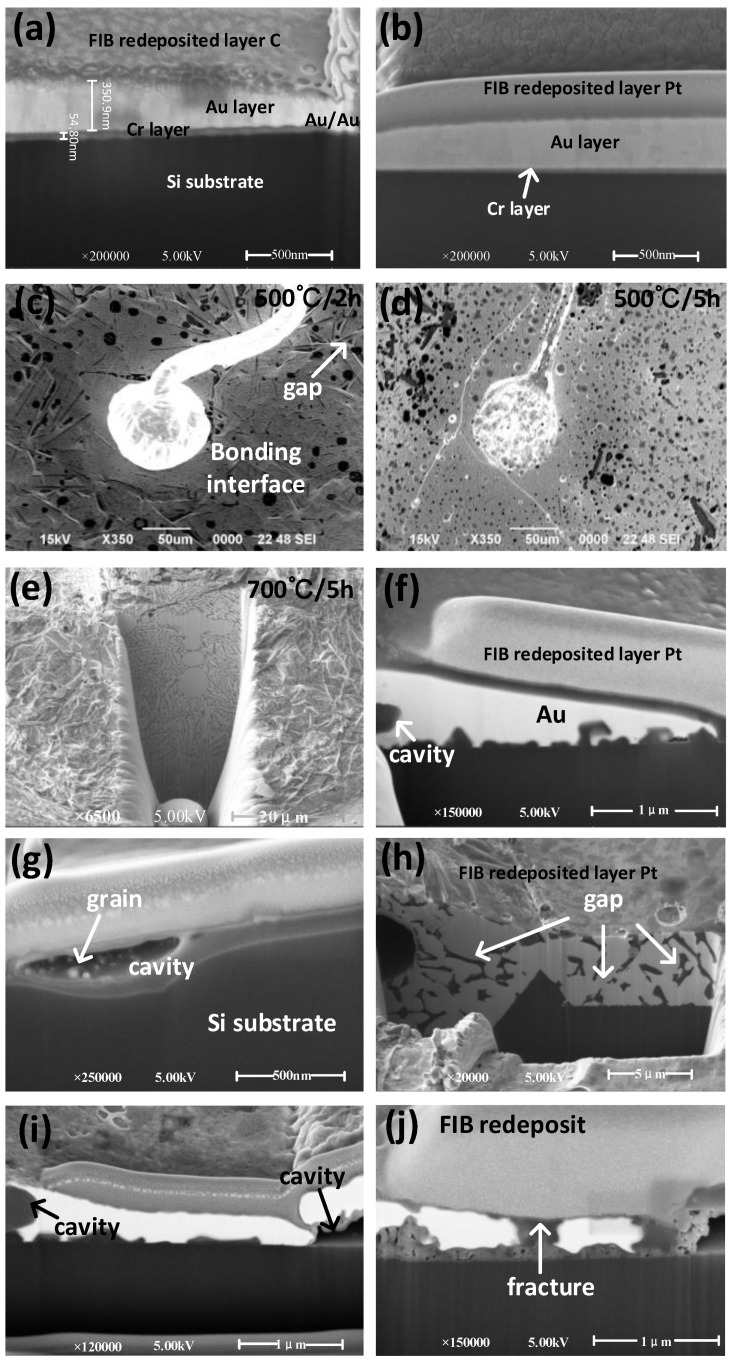
SEM image of FIB cross section cutting of samples covered with a C/Pt (1 µm) electrodeposition layer on gold electrode surface before and after high-temperature annealing as well as the overall SEM image of the uncut bond site after annealing and the SEM image of the FIB cross section cutting of the sample without covered with a C/Pt electrodeposited layer. Overview image of electrode covered with (**a**) C electrodeposition layer and (**b**) Pt electrodeposition layer. Bonding interface image after annealing at (**c**) 500 °C for 2 h and (**d**) 500 °C for 5 h. (**e**) SEM image of bonding interface sectional drawing cut by FIB after annealing at 700 °C for 5 h. SEM image of FIB cut with 300 nm thick electrode annealed at (**f**) 500 °C for 2 h and (**g**) 500 °C for 5 h, SEM image of FIB cut with 400 nm thick electrode annealed at (**h**) 500 °C for 5 h and (**i**) 700 °C for 2 h. (**j**) SEM image of FIB cut with 300 nm thick electrode annealed at 700 °C for 5 h.

**Figure 6 micromachines-13-01603-f006:**
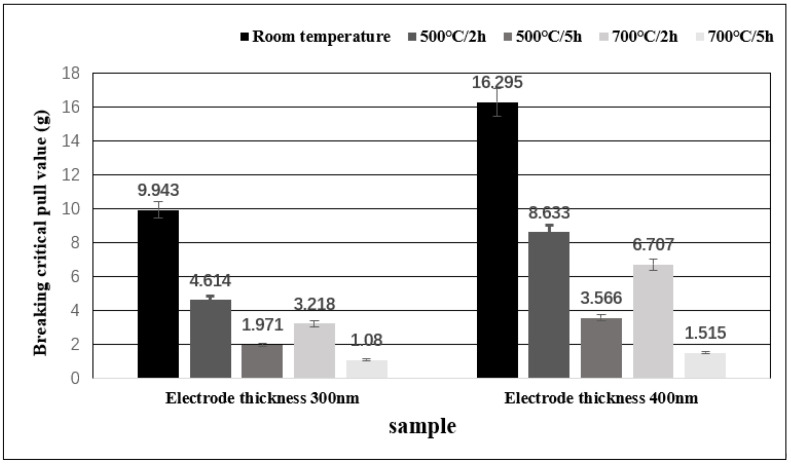
Histogram depicting the critical failure tension of the initial wire bonded samples at room temperature and the heat-treated bonded samples.

**Figure 7 micromachines-13-01603-f007:**
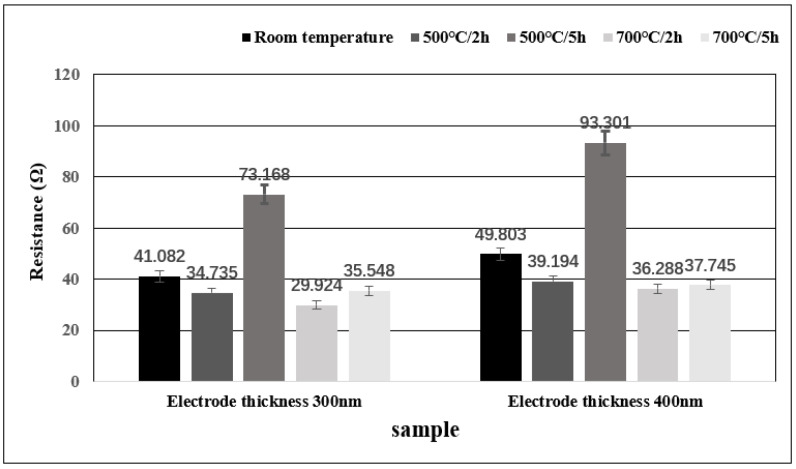
Histogram depicting the resistance change of the initial wire bonding sample at room temperature and after annealing.

**Table 1 micromachines-13-01603-t001:** Characteristics of experimental samples and experimental environment.

Thickness of Thin Metal Layer Cr/Au (nm)	Sample	Temperature (°C)	Time (h)
50/300	S-1	No	0
50/300	S-2	500	2
50/300	S-3	500	5
50/300	S-4	700	2
50/300	S-5	700	5
50/400	S-6	No	0
50/400	S-7	500	2
50/400	S-8	500	5
50/400	S-9	700	2
50/400	S-10	700	5

**Table 2 micromachines-13-01603-t002:** Pull test of different bonding points on the same experimental sample.

Sample and Sample Bonding Point	Thickness of Metal Layer Cr/Au (nm)	Breaking Critical Pull Value (g)	Standard Deviation (g)	Average Pull Value (g)
S1–1	50/300	7.080	--	--
S1–2		9.271	1.549	8.176
S1–3		7.566	1.150	7.973
S1–4		10.341	1.512	8.563
S1–5		14.486	3.115	9.829
S1–6		10.515	2.800	9.943
S2–1	50/400	16.171	--	--
S2–2		15.506	0.470	15.839
S2–3		17.208	0.858	16.295
S3–1	50/400	12.497		
S3–2		13.357	0.607	12.927
S3–3		12.591	0.667	12.815

## Data Availability

Not applicable.

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
