# Peer review of "Au Wire Ball Welding and Its Reliability Test for High-Temperature Environment"

_micromachines, 2022, doi:10.3390/mi13101603_

Round 1

Reviewer 1 Report

In the manuscript, the authors present the reliability of Au wire welding under high-temperature environment. Bonding interface, mechanical strength and electrical resistance are analyzed in detail. This research can provide a significant reference for packaging of high-temperature sensor. I am happy to recommend it for publication and list some suggestions.

1. In Fig. 1, the sample needs further description of the structure. Please provide the size of the sample after dicing and provide the spacing of the electrodes.

2. As shown in Table 2, the test points of two samples are 6 and 3 respectively. Please supplement the data of the second sample to keep consistent with the first sample.

3. In Fig. 5 (a) and Fig. 5 (b), please mark the positions of Cr layer and Au layer, and there are some minor problems with the drawing notes. Please modify it in correct format.

4. In “3. Results and discussion” section, the subtitles need to be adjusted and integrated. Some text descriptions need to be revised to make necessary simplification and result analysis.

5. In “References”, attention should be paid to the format and standardization.

Reviewer 2 Report

The present study reveals how the bond quality of thermosonically bonded Au wires is influenced by the pad metallization thickness (Au pad metallization) and thermal exposure (500°C and 700°C).  Destructive wire pull tests and resistance measurements are employed to determine the bond quality.
Overall the study is thorough but not revealing any new insights and has the scientific merit of a report rather than a journal article.  While the wire pull test reflects the pull force necessary to fracture the wire ABOVE the bond (except for very poor bond quality where a liftout at the bond interface may occur, the typical fracture mode is a wire fracture, as has been wideley discussed in previous publications).
I am therefore wondering why the authors did not perform standardized ball shear tests, too which reflect a bit more the fracture of the bond interface.

In my opinion the present manuscript can only be accepted for publishing with the following major revisions:

1. Add ball shear test results to determine the bond quality and include imaging of the fracture paths.

2. Characterize and discuss better how interdiffusion of the Cr adhesion layer and the Si substrate into the gold metallization (due to high temperature exposure) may change the bond interface.

3. Some Figures need to be improved:

- Fig 3 d & e are not visible, replot it using an approprate plotting program, Fig 3 a-c scalebars are hardly readable.

- Fig 5 : a-j again, the scalebar needs improvement, but overall the selected images are of poor quality, the selected regions of interest are difficult to understand. The labels are often confusing and misleading. What does "slicle" mean? Has edx been performed of the cross sectional bond interfaces before and after anneling to study the potential change of phases? What does "FIB redeposit" mean? A FIB redeposition typically occurrs below a gap and not on top of a structure (the authors maybe want to say" Pt protective layer"?

-Fig 6 and 7 have no error bars

4. What was the annealing atmosphere in the furnace? Vacuum (if so, specify), reducing atmosphere?

Reviewer 3 Report

The manuscript described the efforts that were conducted by authors to investigate the Au wire bonding. However, it seems more like a project report to me instead of a research article. Here are the detailed comments:

1. Page 3 line 103, the authors mentioned about the geometric parameters have impact on bonding quality. But this study did not explore it or even clarify its default parameters in the experiment. In this scenario, it's difficult to determine if the results are universally conclusive or not.

2. Several mechanical tests were performed without details. It is known that it's very tricky to do pull test on Au wire bonding. Contact point, speed rate, and many other factors will affect the final results. It is recommended to state all the details to demonstrate the correctness and repeatability of the experiment.

3. Page 6 line 177, regarding the HT reliability test, there is no surprise that thicker Au always shows better performance. the manuscript does not illustrate the meaning of this test. As long as the metal not melting, HT durability is always there. Only FIB images are not persuasive evidence to support the summary. It is important to quantify the results. In addition, the manuscript just commented good or bad, but the industry need pass or fail. It's common sense that thick Au is preferred, bu expensive. If this work can prove thin Au is worse but still usable, then the value of this work will be revealed.

4. The only interesting thing of the manuscript is on Page 9 line 268. An "outlier" was found not to follow the same trend with others. The 500C/5h increased resistance than other legs. What a pity that the authors did not try their best to demonstrate the accuracy of this test and seek for the root cause for it. The comment made on this phenomenon is "probably due to", which is not rigorous and scientific at all.

Round 2

Reviewer 2 Report

While changes have been made by the authors the images are still not satisfactory for publication.

 Fig 3 d & e not visible axes- previously I recommended to replot the height profile- while the authors have boosted the image and labeled the "step" with the thickness result, they still have ignored my strong recommendation. The axes are still not readable at 100% magnification.

Fig 4 c: the graph is a screen shot the axes haven'T been re-labelled (still in an asian language- thus not compreshensible) --> replotting of the load displacement curve is necessary (with visible axis labels)

Fig 5 SEM images are still not clear; the same images have been used without sginificant improvement. The figure labelling and corresponding captions are inconsistent. Fig 5h has been replaced from "silicle" (oder version) to "gap" although to me it looks like a new phase. Other than what I recommended previously . Fig 5a labels (green) are not visible at all either remove or improve. The scale bars are still not readable --> redraw

In this present form I do not recommend for publication.

Reviewer 3 Report

Thanks for authors response. Most of my comments are answered. Though the answers may not be detailed enough, I would like to let it go in current format.

Author Response

We really appreciate your valuable comments, which are very professional and helpful for improving our manuscript.We will then conduct more in-depth theoretical research and experimental characterization to make this study more thorough and detailed.

Round 3

Reviewer 2 Report

after careful proof reading the manuscript may be accepted